# The effect of conflict on damage to medical facilities in Mariupol, Ukraine: A quasi-experimental study

Danielle N. Poole [1,2]*, Daniel Andersen [1], Nathaniel A. Raymond[1], Jack Parham[1], Caitlin Howarth[1], Oona A. Hathaway [3], Kaveh Khoshnood [1], Yale Humanitarian Research Lab[1]

**1** Epidemiology of Microbial Diseases Department, Yale Humanitarian Research Lab, Yale School of Public Health, New Haven, Connecticut, United States of America, **2** Heidelberg Institute of Global Health, Heidelberg University Hospital, Heidelberg, Germany, **3** Yale Law School, New Haven, Connecticut, United States of America

* danielle.poole@yale.edu

**Data Availability Statement:** A de-identified dataset is available at https://data.humdata.org/dataset/ukraine-mariupol.

## Abstract

Medical facilities are civilian objects specially protected by international humanitarian law. Despite the need for systematic documentation of the effects of war on medical facilities for judiciary accountability, current methods for surveilling damage to protected civilian objects during ongoing armed conflict are insufficient. Satellite imagery damage assessment confers significant possibilities for investigating patterns of war. We leveraged commercially and publicly available satellite imagery and cross-referenced geolocated facility data to conduct a pre-post quasi-experimental study of damage to medical facilities in Mariupol, Ukraine as a result of Russia's invasion. We found that 77% of medical facilities in Mariupol sustained damage during Russia's siege lasting from February 24—May 20, 2022. Facility size was not associated with damage, suggesting that attacks on medical facilities are not random but instead may have been the result of intentional targeting. This is the first cross-referenced pre-post census study of the effects of an ongoing conflict on specially protected medical infrastructure.

## Introduction

Medical facilities are civilian objects that are specially protected by international humanitarian law. Moreover, medical personnel and the wounded and sick must be respected and protected and therefore cannot be intentionally targeted. These protections are enshrined in the 1949 Geneva Conventions [1–3] and the two additional protocols to the Conventions [4, 5]. Serious violations of these protections, including "extensive destruction and appropriation of property" that is protected, may constitute war crimes [1–3].

Despite the protection of international humanitarian law, violence against medical personnel and facilities has been employed in armed conflicts around the world for decades [6–9]. Attacks on health care continue to characterize recent conflicts [10] including the Syrian Civil

**Funding:** This work was supported by the US State Department's Conflict Observatory in the form of a grant (GR122699 to KK, NAR, and DNP) and the Yale School of Public Health Rapid Response Award in the form of an award (to KK, NAR, and DNP). The funders had no role in study design, data collection and analysis, decision to publish, or preparation of the manuscript.

**Competing interests:** The authors have declared that no competing interests exist.

War, [11] the Tigray War in Ethiopia, [12] Israel's Swords of Iron bombardment campaign in Gaza, [13, 14] and the Sudan Civil War [15]. Since the start of Russia's full-scale invasion of Ukraine in 2022, the burden of attacks on medical facilities has been high [16]. Beyond direct harm to health care workers and patients, attacks on medical facilities in Ukraine have severely disrupted access to health care due to security concerns, restricted mobility, broken supply chains, and high rates of absenteeism and health workforce turnover [17, 18]. Quantifying the scale of attacks on medical facilities in Ukraine represents an important step forward toward accountability [19].

Surveillance of attacks on medical facilities [20] is an important tool for accountability and deterrence [21]. Resolutions from the World Health Assembly, the United Nations (UN) Security Council, and the UN General Assembly have reiterated the need to compile data on these violations to protect access to health care in conflict [22–24]. With the mandate to provide leadership on the collection and dissemination of data about attacks on medical facilities, the World Health Organization established the Surveillance System for Attacks on Health Care in 2015. Additional platforms to document attacks on medical facilities are hosted by a limited group of international organizations (e.g., the Armed Conflict Location and Event Data Project, [25] Insecurity Insight, [26] and the Safeguarding Health in Conflict Coalition [10]) as well as local organizations such as the Ukrainian Healthcare Centre (UHC) [27].

Despite the importance of documenting attacks on medical facilities, [8] coordinated and systematic data collection efforts have lagged. Evidence of patterns of attacks on healthcare reported in the peer-reviewed literature is weak, with only 11% of studies involving the collection of quantitative data. [8] Existing public surveillance data is also of insufficient quantity and quality [10]. Although there are reasonable limitations to the responsible sharing of location information in active conflict settings, databases lacking geocoding sufficient for subnational disaggregation preclude independent verification [10]. Analyses of georeferenced databases often fail to corroborate health facility location information, and reliance on such datasets may produce erroneous results [28, 29]. Physical building attributes relevant for understanding probabilistic patterns of attacks, such as building footprints, have, to our knowledge, not been assessed in investigations of damage to medical facilities. Finally, the absence of standardized incident classification and verification across data sources and privileging of particular conflicts and forms of attacks further limit validity [30, 31]. These weaknesses have produced a restricted, non-representative evidence base [32] with a particular dearth of quantitative data. The absence of systematic surveillance of attacks on protected medical facilities during conflict makes it difficult to identify trends, [33] weakens efforts to prevent future attacks, [21, 30] and limits the potential of accountability mechanisms.

Unless attacks are systematically documented, there will continue to be an important gap in knowledge regarding the extent of damage to medical facilities during armed conflict [33]. This gap, if left unaddressed, will have consequences for the provision of medical assistance during ongoing conflict and efforts to reconstruct health systems after armed conflict. Moreover, this gap will undermine efforts to hold alleged perpetrators accountable. Satellite imagery damage classification is considered a "definitive" source of data [34] and is often the only pathway available for empirically detecting and further investigating the effects of conflict on medical facilities [35, 36]. Such remote assessment is increasingly central to the collection of evidence of alleged war crimes and human rights violations, [37, 38] including as evidence in cases before the International Criminal Tribunals for the former Yugoslavia, the International Criminal Court, and the International Court of Justice [39–41]. Today, near real-time, georeferenced damage classification can be performed in nonpermissive conflict environments using publicly available satellite imagery [42].

We leverage commercially and publicly available satellite imagery and medical infrastructure databases to determine the prevalence of attacks on medical facilities during the siege of Mariupol in a pre-post quasi-experimental study. We strengthen the generalizability of our methods by generating a comprehensive, cross-referenced geocoded dataset of damage to all medical facilities within the city of Mariupol during the siege lasting from February 24—May 20, 2022. Further, we examine the relationship between facility size and damage status as an indicator of intentional targeting of medical facilities. Together, these methods generate replicable and reproducible evidence of the impact of the conflict on medical facilities in Mariupol, Ukraine.

## Methods

This study includes a census of damage to medical facilities within the city of Mariupol, located in the Donetsk Oblast of Ukraine, during the siege beginning on February 24 and lasting until May 20, 2022, as part of Russia's invasion of Ukraine. The city of Mariupol has an area of 166 km$^2$. The population of Mariupol has decreased five-fold since the invasion, from 425,000 to 85,000 inhabitants [27]. Prior to the start of the war in 2022, the city had an extensive network of medical care facilities including five city hospitals, one regional hospital, a network of six primary care centers with multiple service points, specialized facilities providing maternal and pediatric care, and private medical facilities [27].

### Study design

This observational, pre-post quasi-experimental study identifies the effects of Russia's invasion on external damage to medical facilities within Mariupol during the siege.

### Ethical considerations

This study is non-human subjects research of publicly and commercially available data. Additional information regarding the ethical, cultural, and scientific considerations specific to inclusivity in global research is included in the (S1 Checklist).

### Data sources

Publicly available, georeferenced medical facility data and satellite imagery were combined to generate an empirical census of damage to medical facilities in the study setting.

**Administrative boundaries.** Shapefiles storing the attribute information of geographic features, including administrative boundaries of Mariupol, were obtained from Natural Earth [43].

**Medical facility validation procedures.** Comprehensive, georeferenced sources of medical facility data vary in quality and completeness [28]. Moreover, these limitations are exacerbated by conflict, and compounded by limited data sharing of medical facilities located in war zones. We used medical facility location data from multiple, publicly available databases including (1) OpenStreetMap (OSM), (2) Google Maps, (3) Wikimapia, and (4) UHC to generate a cross-referenced dataset. The included databases are generated independently of each other. Of note, past examples of geolocated medical facilities have relied on a single data source, although such databases are known to have heterogenous content and quality. Cross-referencing facility locations increases the validity of our findings [29].

First, medical facility polygon and latitude and longitude point layers were extracted from the OSM database using the facility-type attribute. The OSM dataset is a collaborative project designed to create a free and editable geospatial database of the whole world. OSM is one of

the most successful examples of a volunteered geographic information project built by a large user community that employs aerial imagery, GPS devices, and low-tech field maps to verify that OSM is accurate and up to date.

Facilities were included in the study sample if they were categorized as hospitals, clinics, or facilities providing urgent or emergency medical care with a database entry subtype indicating they were a hospital or clinic. This dataset was merged with the geocoordinates of medical facilities documented by UHC [27]. Facilities included in both OSM and UHC databases were considered verified by multiple sources. Facilities included in only OSM or UHC databases were further verified in Google Maps and Wikimapia. The underlying sources of the geolocated entities within Google Maps included publicly available data, licensed third-party data, and data contributed by users [44]. Publicly available and third-party data may be associated with dataset-specific metadata that describe their accuracy and completeness, and, as with OSM, users of Google Maps can flag and report potential errors.

Facilities identified in only one database were excluded. The resulting dataset included 73 facilities identified in two or more databases. Latitude and longitude were recorded for each cross-referenced medical facility.

**Facility characteristics.**   Building footprint and height were derived from OpenStreetMap [45]. For facilities comprised of multiple buildings, polygons were grouped into a complex. Area was calculated by summing the area of the building polygons for each medical facility. Building height was calculated by multiplying the highest floor by 3.9 meters [46]. Google Maps and Yandex were used to determine the number of floors for polygons missing this information in OSM. Volume was calculated for each polygon and summed to generate the medical facility volume.

## Damage classification

Damage to medical facilities is an observable form of violence against medical facilities [10]. Every medical facility identified and verified through open source documentation was assessed via satellite imagery analysis to establish a baseline structural assessment. The very high resolution imagery used to support this investigation was unclassified, commercially available imagery captured by Maxar Technologies, Planet Labs, and other commercial satellite imagery suppliers provided by the US Department of State to the Humanitarian Research Lab at Yale University School of Public Health. An overview of the technical specifications of the images included in this study is provided in **Table 1**.

**Table 1. Satellite imagery technical specifications.**

| Date and timestamp | Provider | Ground sample distance (cm) | Cloud cover (%) | Off nadir angle (˚) |
|---|---|---|---|---|
| 2021-06-21 08:48 AM UTC | WV02 | 50 | 0 | 18.1226 |
| 2022-03-13 11:14 AM UTC | WV01 | 50 | 11.81 | 31.0658 |
| 2022-03-14 08:53 AM UTC | WV02 | 50 | 10.59 | 32.9965 |
| 2022-03-29 08:29 AM UTC | WV03_VNIR | 38 | 0 | 27.1867 |
| 2022-03-29 08:29 AM UTC | WV03_VNIR | 38 | 0 | 27.1867 |
| 2022-04-03 08:50 AM UTC | GE01 | 50 | 5.58 | 32.2946 |
| 2022-04-28 05:32 AM UTC | Skysat | 50 | 30 | NA |
| 2022-05-06 08:50 AM UTC | GE01 | 49 | 4.04 | 24.5753 |
| 2022-05-08 08:52 AM UTC | WV03_VNIR | 42 | 0 | 32.632 |
| 2022-05-12 08:18 AM UTC | WV03_VNIR | 38 | 0 | 26.4653 |
| 2022-07-13 11:46 AM UTC | WV01 | 50 | 4.24 | 37.0499 |

**Table 2. Damage classification.**

|  | Definition |
|---|---|
| *No visible damage/Possibly damaged* | The building appears to have complete structural integrity; the walls remain standing; the roof is undamaged. |
|  | Or, uncertain interpretation due to image quality; presence of possible damage proxies like small traces of debris/rubble or sand deposits around building. Building surrounded by damaged/destroyed buildings. |
| *Damaged* | Including both minor and major damage. |

Adapted from IWG-SEM [47]

Multi-temporal change detection between pre- and post-siege imagery was conducted independently by a minimum of two geospatial analysts. Disagreements were resolved via consultation with a third analyst.

First, pre-siege imagery of all facilities taken prior to the start of the siege on February 24, 2022 was assessed to establish a baseline. A second damage assessment of each facility was conducted on imagery dated after the start of the siege until the end of the siege. Post-siege images included imagery taken throughout the study period in order to identify any incident of damage, given the binary damage classification employed. Facilities that had not sustained damage earlier in the siege were assessed on or before May 20, 2022 and the nearest following date with clear imagery. Indicators of damage to infrastructure include changes in feature coloration, texture, and pattern as seen from above. Spatial resolution between 32 and 50 cm and temporal resolution in this imagery allowed analysts to assess changes in infrastructure and the natural environment, both of which may visibly reveal damage caused by heavy weapons and some small arms commonly utilized in armed conflict.

Damage visible in satellite imagery was used to classify medical facilities as sustaining: 1) no visible damage/possible damage; and 2) damage. These categories were defined by the damage assessment scale developed by the International Working Group on Satellite-based Emergency Mapping (IWG-SEM) and are described in **Table 2** [47].

While numerous building damage assessments are used in natural disaster and conflict analyses, [48] the IWG-SEM represents international consensus guidance for aerial assessments in emergency settings. For the purpose of estimating the proportion of medical facilities that were subject to damage during the siege, no distinction was made between minor and major damage classification in this study. Distinction between IWG-SEM damage severity classifications was not always feasible due to image quality, as shown in **Fig 1**. Thus, a binary damage classification was applied to all images for consistency.

## Statistical analysis

This study aimed to conduct a census of damage to all medical facilities in the study setting. The study variables were analyzed using frequency, percentage, mean, median, and interquartile range (IQR). Logistic regression models quantified associations between categorical damage and building characteristics. All analyses were conducted in Stata version 17.0 [49].

## Results

A total of 169 points and polygons indicated to have a medical function were captured across the four datasets. Ninety-six instances were excluded after removing duplicates, non-medical facilities, and facilities that were not cross-referenced in a minimum of two datasets.

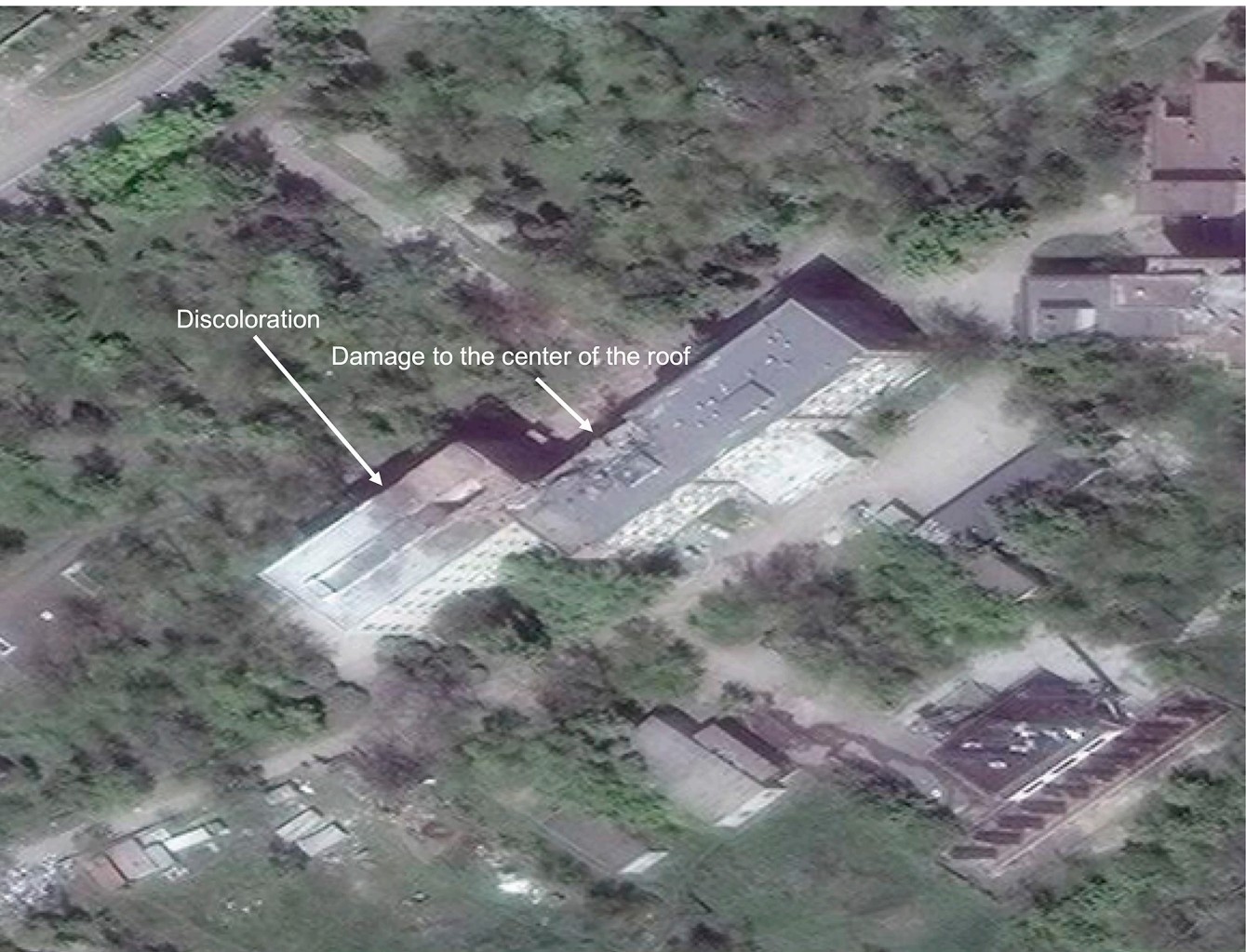

Discoloration

Damage to the center of the roof

**Fig 1. Damage of indeterminate severity.** *Discoloration along the western portion of the roof and clear signs of damage in the center of the roof are visible. However, the severity of the damage, assessed as the structural integrity of the roof, is indeterminate. Due to the clear signs of damage, this structure would be classified as "damaged" in the adapted IWG-SEM schema used in this study.* Republished from Maxar Technologies Inc. under a CC BY license, with permission from the US Department of State, original copyright ©2022 MAXAR. Damage assessments were repeated on at least one image during or immediately after the siege.

### Medical facility damage census

A total of 73 medical facilities were geolocated and cross-referenced within the city of Mariupol. All 73 medical facilities included in this study were confirmed intact at baseline. The percent agreement for damage classifications performed for this study was 87%.

Over three-quarters (n = 56, 77%) of medical facilities sustained damage during the study period. **Fig 2** presents an example of commercially available satellite imagery remote detection of damage to a medical facility [50].

### Facility characteristics

Medical facility characteristics, derived from OSM, are presented in **Table 3**. The median building was 10,819 cubic meters in volume, with a footprint area of 1,014 meters squared, and a median height of three floors (or, approximately 12 meters). Variation in facility size was

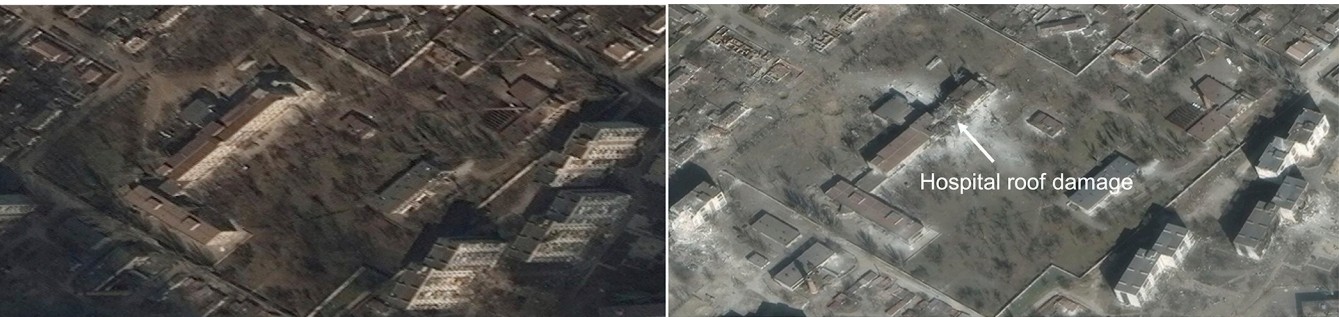

**Fig 2. Damage to the Mariupol Maternity Hospital consistent with direct impact from bombardment.** *Mariupol Maternity Hospital on March 18, 2022 (left) and March 29, 2022 (right).* Republished from Maxar Technologies Inc. under a CC BY license, with permission from the US Department of State, original copyright ©2022 MAXAR.

notable: the facility with the largest footprint was 2.4-fold larger than the facility with the smallest footprint. The volume of the largest facility was 4.4-fold greater relative to the facility with the least volume.

There was no significant difference in the building size, including the overall volume, height, and footprint, of medical facilities that sustained damage and those that did not, as presented in **Table 3**.

The lack of difference in the size of facilities that sustained damage is suggestive of intentional targeting of medical facilities, which would constitute a war crime, as random saturation of artillery fire or unintentional damage would be expected to hit larger facilities at a higher frequency.

## Discussion

We present independent evidence of widespread damage to cross-referenced medical facilities during the besiegement of Mariupol. This study also demonstrates the feasibility of generating reproducible and replicable [51] estimates of the effects of conflict on medical facilities using commercially and publicly available data sources, a previously noted gap in the literature analyzing attacks on healthcare [52]. These methods represent an opportunity to strengthen existing surveillance systems of attacks on medical facilities.

We found a preponderance of evidence of damage to medical facilities, which are specially protected civilian objects, in the city of Mariupol. Each incident of an attack on a medical facility may represent an individual and distinct violation of international humanitarian law and therefore a possible war crime [1–5]. The percentage of damage identified in this study is consistent with previous reports that 80% of the health care infrastructure in Mariupol has been destroyed [27]. Our results are also aligned with those of the UHC, which reported damage or destruction to 77% of facilities in Mariupol [27]. The similar proportions of observed damage identified by satellite imagery, considered "definitive," [34] and evidence drawing from open data sources suggests the validity of the approach used in this investigation. Furthermore,

**Table 3. Medical facility characteristics and associated damage status.**

| Characteristics | Total (n = 73) | Damaged (n = 56) | UOR (95% CI) | p-value |
|---|---|---|---|---|
| *Building height median (meters, IQR)* | 12 (8–16) | 12 (8–18) | 0.00 (-0.05–0.04) | p<0.868 |
| *Building area median (square meters, IQR)* | 1,014 (733–1,562) | 1,056 (647–1,580) | 0.00 (0.00–0.00) | p<0.728 |
| *Building volume median (cubed meters, IQR)* | 10,819 (4,663–19,635) | 10,431 (4,446–19,332) | 0.00 (0.00–0.00) | p<0.912 |

satellite imagery damage assessment of cross-referenced facilities may represent an opportunity to standardize reporting for direct comparisons within and across conflicts.

Over eighty-percent of attacks on medical facilities in Ukraine are reported to have been performed with heavy weaponry, typically trained on the largest structures [16, 53]. In the absence of other risk factors, larger buildings would be expected to be subject to more attacks. However, our findings demonstrate that damage was not associated with multiple measures of building size; larger buildings did not have a higher likelihood of being damaged. There are two possible explanations for this finding. First, this pattern may demonstrate intentional targeting of medical facilities. Alternatively, near comprehensive infrastructure destruction may have generated widespread, indiscriminate damage. Regardless, these findings demonstrate a failure to protect civilian objects from attack.

This study identified attacks on medical facilities occurring when Mariupol was under siege by Russia [54]. The Armed Forces of the Russian Federation have been identified as the actor associated with aerial strikes that would be expected to result in the damage to medical facilities observed in this study [55]. To our knowledge, there is no evidence that Ukrainian troops conducted airstrikes on any medical facility in Mariupol during the study period. Beyond the broad linkage of damage, two years of research by human rights organizations has identified the specific units of the Armed Forces of the Russian Federation responsible for the extensive destruction of Mariupol [55].

Medical facility locations are increasingly maintained in online databases with information such as building latitude and longitude, footprint polygons, and imagery of signage [56, 57]. The locations of all medical facilities included in this study are available to the public, including to the Armed Forces of the Russian Federation, in a minimum of two databases. Intentional targeting of medical facilities would clearly violate international humanitarian law and would constitute war crimes. The only exception to this rule is in cases where the forces targeting medical facilities have clear reason to believe that those facilities were used for non-medical military purposes. One such claim has been found baseless [55]. Even if medical facilities were used for military purposes, the prohibition on causing excessive harm to civilians and civilian objects would in most cases prohibit strikes. Moreover, the requirement that armed forces take precautions to avoid damaging medical facilities means that the extensive damage to medical facilities in Mariupol would violate international humanitarian law regardless of whether there was a specific intent to target medical facilities [58].

## Strengths

A major strength of our study is the pre-post quasi-experimental study design used to estimate the effects of Russia's siege on medical facility infrastructure by applying reproducible and replicable methods to publicly available and cross-referenced data sources, accounting for building features relevant to targeting. The resulting scientifically rigorous findings are expected to foster a continuum of knowledge, evidence, and practice that can support war crime accountability [33]. Moreover, the analysis of publicly available data for the generation of evidence is available to all actors and in a variety of settings, from international to domestic operational theaters to courtrooms and diplomatic settings.

## Limitations

This study has several limitations. First, damage assessments conducted primarily using satellite imagery are limited to visible features and may not always show damage to a structure's sides and interior. Furthermore, the binary classification of damage used in this study precludes analyses of damage severity and multiple incidents of attacks on the same facilities, both

of which may provide further evidence of targeting. Thus, this study provides a conservative estimate of damage. Combining this data with ground-level assessments and additional sensors, including medical facility type and functionality indicators, would further illuminate the scope of damage. Nevertheless, cyber forensics capacity, including the analysis of satellite imagery, represents an important source of evidence for national and international criminal justice [37, 59]. Second, this study does not assess whether the identified medical facilities may have been used for non-medical military purposes, thus becoming a lawful military object, or whether there may have been adjacent buildings that constituted lawful military objects that may have been the true targets of the attacks. Yet, even if Russia's armed forces had military objectives in conducting the attacks, the special protected status of medical facilities should have resulted in the application of the principles of proportionality and precaution. The international humanitarian law principle of proportionality–which prohibits attacks against military objects that are expected to cause incidental harm to civilians or civilian objects that would be excessive in relation to the concrete and direct military advantage anticipated–likely would have rendered the attacks unlawful [58].

## A way forward for war crimes accountability

Even recognizing these limitations, this study offers an important way forward for accountability for violations of international humanitarian law—and thus, it is hoped, for deterring future violations. A key difficulty in war crimes prosecutions has long been evidence collection. During ongoing conflict, evidence can be difficult, if not impossible, to gather—and doing so can pose serious dangers to investigators. Moreover, given limited and sporadic access to the site of violations, it can often be challenging to pinpoint when a violation took place: information that is necessary to building evidence that can be used in a criminal prosecution. This study demonstrates that reproducible and replicable evidence of potential violations of international humanitarian law can be gathered using publicly available satellite imagery together with other public data sources. In doing so, it offers an important solution to longstanding problems in providing accountability for violations of the law.

It is important to recognize that evidence of potential violations of international humanitarian law is a necessary, but not sufficient, step towards war crime accountability. Most important, cases must be brought against perpetrators–either in international courts, such as the International Criminal Court, or in domestic courts [60]. Moreover, for such evidence to be used effectively, courts and those practicing before them must develop expertise in and procedures for gathering, assessing, and preserving these new forms of evidence. For example, integration of methods demonstrating systematic damage to protected civilian objects, such as medical facilities, should be incorporated into the International Criminal Court Rules of Procedure and Evidence to ensure the generation and preservation of admissible evidence [41].

This study fills a high-priority gap in surveillance [52] and demonstrates a technique that may be adapted to a variety of contexts, including local munitions profiles and the availability of geolocated medical facility data. Future research should seek to expand the scalability of these methods by testing the validation of supervised automation and to document other legal violations, including war crimes beyond the targeting of medical facilities and violations of international human rights law, accountability for which often suffers from the same evidentiary hurdles. Future satellite imagery analysis of aerial spray may also be used to detect the source of attacks, which represents an important contribution to attribution. Finally, future research may include developing real-time remote indictors of medical facility functioning to serve operations in ongoing conflict.

## Conclusions

To our knowledge, this is the first evidence of widespread damage to cross-referenced facilities in a pre-post quasi-experimental study of the effects of war on urban medical infrastructure. As such, the reproducible methods, incorporating publicly and commercially available data, have the potential to make substantial contributions to the international justice system by providing real-time estimates of damage to specially protected civilian objects. By combining the strengths of satellite imagery analysis in nonpermissive environments with publicly available data sources of medical facility infrastructure, this study demonstrates the power of data amalgamation for creating outputs that were, until recently, inaccessible. The findings presented here support the existing evidentiary basis that attacks on Ukraine's health care system constitute war crimes [61]. However, because the impact of this work is contingent upon its use in international justice mechanisms, this study also represents a call for action to clarify the forensic standards for remote documentation of war crimes and expand expertise in collecting and assessing such evidence.

## Supporting information

**S1 Checklist. Inclusivity in global research.**
(DOCX)

## Acknowledgments

The authors are grateful for the contributions of anonymous Ukrainian colleagues. This study is a product of the members of the Yale Humanitarian Research Lab who are leading the advancement of war crime documentation. This independent study builds upon ongoing work supported by the US State Department and Conflict Observatory consortium. This study does not necessarily reflect the views of the United States Government.

## Author Contributions

**Conceptualization:** Danielle N. Poole, Nathaniel A. Raymond, Caitlin Howarth, Kaveh Khoshnood.

**Data curation:** Danielle N. Poole, Daniel Andersen, Jack Parham.

**Formal analysis:** Danielle N. Poole, Nathaniel A. Raymond, Oona A. Hathaway.

**Funding acquisition:** Danielle N. Poole, Nathaniel A. Raymond, Kaveh Khoshnood.

**Investigation:** Danielle N. Poole, Daniel Andersen, Nathaniel A. Raymond, Caitlin Howarth, Kaveh Khoshnood.

**Methodology:** Danielle N. Poole, Daniel Andersen, Nathaniel A. Raymond, Caitlin Howarth, Kaveh Khoshnood.

**Project administration:** Danielle N. Poole, Caitlin Howarth.

**Supervision:** Danielle N. Poole, Nathaniel A. Raymond.

**Visualization:** Daniel Andersen.

**Writing – original draft:** Danielle N. Poole.

**Writing – review & editing:** Danielle N. Poole, Daniel Andersen, Nathaniel A. Raymond, Jack Parham, Caitlin Howarth, Oona A. Hathaway, Kaveh Khoshnood.

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
