## [Decision Letter · Decision Letter 0]

25 Jun 2024

PGPH-D-24-00764

The effect of conflict on medical facilities in Mariupol, Ukraine: a quasi-experimental study

Dear Dr. Poole,

Thank you for submitting your manuscript to PLOS Global Public Health. After careful consideration, we feel that it has merit but does not fully meet PLOS Global Public Health’s publication criteria as it currently stands. Therefore, we invite you to submit a revised version of the manuscript that addresses the points raised during the review process.

We look forward to receiving your revised manuscript.

Kind regards,

Dr. Ruwan Ratnayake 

Academic Editor

PLOS Global Public Health and LSHTM

Journal Requirements:

2. We ask that a manuscript source file is provided at Revision. Please upload your manuscript file as a .doc, .docx, .rtf or .tex.

Additional Editor Comments (if provided):

Thank you for sharing your important analysis.

Please pay particular attention to the comments by reviewer 2 on:

- The risk of misattribution of ACCLED data and improving its alignment with building damage. Do you need this analysis, given the lack of temporal alignment?

- Relevance of this particular analysis to IHL arguments and what further work needs to be done to make these links more robust.

And reviewer 1 on:

- Distinguishing between conflict events and direct attacks

- Attribution of damage to Ukrainian and Russian Forces respectively

- More detail on methods for joining ACCLED and satellite data

- Tempering expectations of attribution links between damage and perpetrator based on your findings, and what future research can be done to improve this.

Though this may seem obvious, please also explain in the introduction how physical damage to health facilities likely results in reduced capacity, poor access, fear for health workers to come to work, etc. I am also interested in the lack of involvement of Ukrainian co-investigators from Mariupal, and the potential insights into the methods and results which may have been missed. Did the team attempt to gain local knowledge, and are there Ukranian authors left out for reasons of their own safety?

Feel free to mention where you disagree with the comments or feel that you have a better suggestion than what is offered.

Reviewers' comments:

Reviewer's Responses to Questions

**Comments to the Author**

1. Does this manuscript meet PLOS Global Public Health’s publication criteria? Is the manuscript technically sound, and do the data support the conclusions? The manuscript must describe methodologically and ethically rigorous research with conclusions that are appropriately drawn based on the data presented.

Reviewer #1: Partly

Reviewer #2: Yes

2. Has the statistical analysis been performed appropriately and rigorously?

Reviewer #1: No

Reviewer #2: Yes

3. Have the authors made all data underlying the findings in their manuscript fully available (please refer to the Data Availability Statement at the start of the manuscript PDF file)?

Reviewer #1: Yes

Reviewer #2: No

4. Is the manuscript presented in an intelligible fashion and written in standard English?

Reviewer #1: Yes

Reviewer #2: Yes

5. Review Comments to the Author

Reviewer #1: Overall feedback (more details below):

It is notable and unfortunate that no colleagues from Ukrainian institutions were included as authors in this research, especially given how local insight could have enriched interpretation of findings.

While the paper outlines a quality, reproducible method for assessing damage to health infrastructure during conflict, it seems to overstate its significance in building evidence to hold perpetrators accountable. Documenting damage to health facilities as a result of general conflict without strong causal/attribution links is distant from providing sufficient evidence to hold perpetrators accountable for IHL violations. This is especially true when both sides of the conflict are listed as the main actors for conflict events causing damage to health facilities.

There needs to be a clear distinction between the following concepts and appropriate application throughout the manuscript, as both are conflated in the paper:

General conflict events which results in collateral damage to health facilities vs. direct attacks on health facilities. As insufficient detail is provided in the methods section on how attacks are matched with facilities, it is difficult to know which is addressed. Consistent language should be used on whether they are conflict events or direct attacks.

Documenting attacks on health facilities vs. estimating impact of conflict events/attacks on health facilities

Major bias introduced by omitting any descriptive analysis or comments on the fact that 18 of the 56 (32%) damaged facilities list the Ukrainian military as the main actor. The paper implies that all attacks and damage are the result of Russian Federation attacks.

This paper overlooks key literature on the topic, including:

o https://phr.org/our-work/resources/russias-assault-on-ukraines-health-care-system/

o https://conflictandhealth.biomedcentral.com/articles/10.1186/s13031-023-00557-2

o and does not give appropriate attention to a third (though it’s referenced in 32) https://drive.google.com/file/d/1tJyyae_9eBF_IfkPiDb5i9D2Bqb- 2n5w/view . This paper seems to be a secondary approach to UHC’s 2022 report on the same topic. As such, this should be referenced more clearly and prominently in the background or discussion.

Methods, results, and discussion should be strengthened.

o While the background focuses on weaknesses in documenting attacks on health facilities, a weak methods section makes it unclear whether this paper provides an approach that improves upon existing surveillance activities.

o Detail is needed on cleaning of the ACLED dataset and how facilities and conflict events are matched. Without this, it is difficult to ascertain the likelihood that the damage is attributable to a specific conflict event/actor.

o Results section doesn’t provide descriptive summaries of conflict events and cherry picks certain messages (i.e. highlighting Russian attacks without acknowledging Ukrainian attacks matched with damaged facilities)

o Discussion section does not explore or address existing literature on the topic or possible alternative explanations for findings. While understandably this paper can’t make the link between damage and the level of evidence for an IHL conviction, the paper should be framed as a building block and provide feedback on how future research can build on this step to move towards holding perpetrators accountable. Currently, it reads as though the paper on its own will lead towards more perpetrator accountability.

Referencing:

o Many references are not accurately represented or statements in the manuscript are not backed up by the provided references. (Examples provided below)

o Many references are missing links, which are especially important for grey literature. Ex. Reference 32 does not produce any immediate results in a Google search with that exact title.

Data sources - Health facility list

o Was any effort made to confirm the functionality of the health facilities prior to the conflict events? Evidence from other settings suggests there is often a large exodus of health care workforce when conflict begins.

o Was any effort made to confirm the accuracy of medical facility location data? Publicly available data on this is known to be wildly incomplete/inaccurate in other conflict settings.

o Doesn’t clearly define a health facility for the purposes of this study or discuss how different datasets might classify facilities differently. For example, UHC considers the 12 separate outpatient points in 8 communities that make up the Mariupol PHC as separate facilities. Does OSM do the same?

o Would be helpful to state how each health facility dataset sources its data to ensure Wikimapia does not source data from OSM/Google for example. This would lead to invalid verification by multiple sources.

Specific feedback:

The title should be made more specific, as its current phrasing implies a more comprehensive exploration of effects, while this study only examines external damage to medical facility physical infrastructure during the conflict period.

The validity of the following statement is questionable: “This is the first geographically comprehensive pre-post study of the effects of an ongoing conflict on specially protected medical infrastructure.” (lines 35-36). It is unclear how this is different from previously published studies other than using remote sensing data: https://conflictandhealth.biomedcentral.com/articles/10.1186/s13031-023-00557-2 . And geographically comprehensive to Mariupol?

Line 47 – Need to be clear on whether this clause refers to Ukraine or globally. If globally, which it seems to be, the provided reference does not apply. There are more appropriate references, including the SHCC report.

Lines 48-56 – PHR has done one of the more robust efforts to document attacks on health care in Syria and Ukraine, which should be included here, especially as you reference several papers pertaining to Syria in the background.

Line 58-60 – This statement is misleading. The topic sentence focuses on attacks on medical facilities, but the two papers referenced address grave violations against children (not including attacks on health) and attacks on education. The following sentences then jump back to statements related to attacks on health. This should be rewritten for clarity and accuracy.

Lines 57-68 – Conflation between documenting attacks and estimating the effects of attacks on health. This should be clearly separated.

Line 66 - Has grammatical errors and doesn’t make sense.

Line 110-111 – This should be made more specific (along with the title) to specify that this study is on the effects of conflict on external medical facility physical infrastructure.

Lines 129-131 – Missing references - “Of note, past examples of geolocated medical facilities have relied on a single data source, although such databases are known to have heterogenous content and quality.”

Lines 131-132 – Increases likelihood for improved validity of findings. Cross-referencing bad data, doesn’t lead to more valid findings.

Lines 154-160 – Is there more than one polygon for each facility? Why?

Line 164 – What was the result of the baseline structural assessment? Were facilities classified as damaged dropped?

Line 196 – Narrative references IWG-SEM was used, but table references an adapted version of IWG-ES. Why?

Lines 200-202 - Unclear why a distinction wasn’t made between minor and major damage, as one could still calculate the proportion of facilities damaged from minor/major classifications. Especially as the paper is framed around documenting IHL violations to hold actors accountable, being able to categorize whether the damage is superficial or affects health service functionality seems very important.

Lines 202-214 – In the binary system this was classified as not damaged due to insufficient image quality? It is unclear from the example how you distinguish between minor damage and indeterminate damage which would land in different binary classifications.

Lines 222-230 – This leaves a lot to be desired in terms of whether the ACLED data was cleaned and the parameters of the joins. More is needed on how the ACLED data was matched to medical facility polygons. Was the dataset filtered based on location accuracy scores in the ACLED dataset? Were event types filtered out, such as riots/civil unrest, where a mechanism for damage from military attacks would not be present? Was there a cutoff for maximum distance of a conflict event to a health facility? Date range? etc.

Line 250 – Was any effort made to confirm the functionality of the health facilities at the start of the conflict?

Lines 274-277 – Is there a reference to support this bold statement? There seem to be other possible explanations. The spread of facility sizes is not that large, so is there evidence to support the assumption that larger facilities would be more likely to be hit? It seems more likely if comparing facilities 100sqm to 3000sqm, but actual facility footprints don’t seem to vary dramatically. There are presumably more smaller facilities, so stratification would be helpful to know whether the n for large facilities is sufficient to compare. Is there evidence from other studies to suggest that one would expect there to be variation in damage sustained based on these modest size differences? Lots of other explanations seem possible…proximity to strategic targets, large proportion of the area conflict affected, etc..

Lines 280-283 – these numbers don’t sum to 46. Also, more description on what matched means (should be clarified in the methods) is essential to understand the implications of being matched with a conflict event.

Lines 283-286 – Would be helpful to see a better descriptive summary of the characteristics of matched conflict events. What proportion of conflict events were attributed to the Russian Federation? If not the Russian military, then which actor was identified? What was the breakdown by conflict event type? If the Ukrainian military is responsible for one third of the conflict events matched to damaged health facilities, how does this influence the results, discussion, and conclusions?

Lines 292-293 – This statement, like the title of the paper, should be tempered. While the effects of conflict on health facilities can be wide ranging (staffing, service availability, service use, facility functionality, supply chain, etc.), this paper is only able to identify whether any external damage to a facility occurred, not including severity of damage which might further implicate functionality effects. Furthermore, the attribution links between conflict events and damage are weak, especially in the presented framing of IHL accountability.

Line 299 – Typo in reference. Also, the sentence indicates the referenced sources cannot be compared due to methodological differences, but reference 43 simply pulls a number from a news article quoting a politician. There are no methods or data presented behind this statement.

Lines 297-299 – More unpacking of the differences between the studies would be helpful in understanding how this approach is an improvement on the previous estimates and the implications for future research.

Lines 303-305 – Again, would be helpful to have a descriptive summary of the actors and weaponry associated with the included attacks somewhere in the paper.

Lines 306-307 – This statement seems highly susceptible to reporting bias, i.e. heavy weaponry and damage at large facilities would be more likely to be detected and reported than small arms attacks on small facilities. This also cites a system that was highlighted as insufficient in the background section. These, and other considerations, should be noted.

Line 329 – Would the study have been possible without the commercial data from the State Department?

Lines 327-331 – This statement seems far too strong for the evidence presented in the paper. Very weak attribution links are presented and this does not address any of the major hurdles to holding IHL perpetrators accountable.

Lines 331-335 – Isn’t making the evidence available to all parties requisite in trials?

Reviewer #2: Review of PGPH-D-24-00764 for PLoS Global Health

Gregg Greenough

General Comments:

I appreciated the opportunity to review this submission.

The paper presents itself as a quasi-experimental study of pre- and post- invasion analysis of attacks on Mariupol, Ukraine medical facilities, measured as building damage, during the three month siege by Russian forces in 2022. They derive medical facilities from multiple cross-referenced open-sourced and government data sets, create building geolocated polygon footprints, and use satellite imagery with routine classification analysis with an accepted methodology to quantify prevalence of damage and discuss these findings within the principles and framework of IHL.

In general, quasi-experimental designs, by definition, help illuminate causation and from the spatial and temporal standpoint, the paper accomplishes that. Where it falls short is attempting to relate events from the ACLED database with damage without presenting any actual time/place analysis of that data, at which point it detracts from the study design and purpose. Please see my comments below for more specifics – I think it can be done in a way that compliments and enforces the paper’s focus.

The study also successfully argues for more robust infrastructure damage/medical attack surveillance systems upon which they depend, although that is discussed in the Introduction and is not mentioned in the Discussion. But the point is well-taken.

A few other general comments:

1. The derivation of the facility dataset seems reasonable and thoughtful. In fact, the researchers have done due diligence to minimize data gaps that frequently occur in places with on-going conflict.

2. Despite the conservative definition of the binary damage classifications (well-explained in the Limitations), the amount of damage was disturbingly large and that fact alone renders this study valuable in raising IHL allegations.

3. I appreciate the fact that the indiscriminate pattern of damage is brought out in both the Results and Discussion. The strength of the paper highlights what is often missed – not that Russian forces were targeting medical facilities (they may well have), but their obligation under IHL is to not be indiscriminate and not to be disproportionate when their obligation is protecting civilians and civilian infrastructure. What that in mind, I would suggest also making a proportionality argument given the amount of destruction of medical infrastructure in a short time (demonstrated in the analysis), especially, as you later note, realizing that damage is undercounted and underappreciated from your conservative imagery classification.

Specific Comments:

The authors provide a compelling introduction and rationale defining the need for the study and its role in providing additional evidence in a milieu as fraught as the Mariupol siege.

Nonetheless, the authors should make it clear that this study is still a ‘one off’ point in time analysis that has the potential to add to a larger body of evidence if and, hopefully, when it’s needed for international legal bodies to confer.

1. Line 66 – confusing sentence – change perhaps to ‘absence of impact estimations’?

2. Small change: Ref 36 should be “International Working Group” (not Interagency) – please provide the link to public documents, in this case: https://www.un-spider.org/sites/default/files/IWG_SEM_Guidelines_Building%20Damage%20Assessment_v1.0.pdf.

3. Methods –

a. Thanks for introducing us to Natural Earth! Appreciate the need and use for multi-scaled data sets.

b. Agree with using multiple facility datasets and cross-referencing them. This adds strength to the paper and mitigates the limitation of ‘completeness’ in an active conflict zone.

c. Line 102 – should be ‘km2’

d. Please address any potential types of error in the process of deriving facility polygons.

e. For clarity – the building polygons consisted of the footprint of the building only and not the land that it was on, yes? Also, when Google Maps and Yandex were used to determine number of floors, was this Street View? If so, address any potential sources of error with this approach.

f. Were the two imagery analysts classifying independently? I assumed so, but please state.

g. What was the utility of classifying any imagery with significant cloud cover during the siege (since you have other imagery available without cloud distortion during that time)?

h. Analysis – the Clopper-Pearson interval is used appropriately here for the study design of independent binomial outcomes. The authors should state what statistical package they used for calculations.

4. The authors should consider combining Tables 3 and 4 since they describe the same data outputs.

5. The ACLED event information is presented without actual data to demonstrate the connection with building damage. I think you need to provide something here. For instance, when you state the ACLED events were ‘geospatially matched’ (line 280), do you mean they occurred in close proximity to the buildings? While you wisely stop short of directly inferring causality without an analysis (since the ACLED data is not temporally matched), it seems like it would be helpful to clarify what ‘geospatially matched’ actually implies. Also, can the ACLED events be temporally related to the building damage (my fear is that your temporal resolution may not be granular enough)? If you can use a proximity distance measure (event to building) and timestamp of the event to the time of damage, this would be more compelling. The study has shown that considerable damage was done to medical facilities. While it’s probably safe to say that Ukrainian forces were not attacking their own medical facilities during this period, demonstrating that Russian forces inflicting events in time and space during a period of siege would deflect any counter-argument and place the spotlight more squarely on their IHL violations.

6. Conclusions. There have been recent studies in the literature using remote sensed imagery on medical facility infrastructure already (Gaza, for example), so the point here should be rephrased to explain how the use of remotely sensed imagery has value in this particular scenario (urban, siege, time-dependent) and how its use is adding validity to the IHL arguments.

6. PLOS authors have the option to publish the peer review history of their article (what does this mean?). If published, this will include your full peer review and any attached files.

**Do you want your identity to be public for this peer review?** For information about this choice, including consent withdrawal, please see our Privacy Policy.

Reviewer #1: No

Reviewer #2: No

---

## [Decision Letter · Decision Letter 1]

28 Oct 2024

The effect of conflict on damage to medical facilities in Mariupol, Ukraine: a quasi-experimental study

PGPH-D-24-00764R1

Dear Dr. Poole,

We are pleased to inform you that your manuscript 'The effect of conflict on damage to medical facilities in Mariupol, Ukraine: a quasi-experimental study' has been provisionally accepted for publication in PLOS Global Public Health.

Best regards,

Ruwan Ratnayake, PhD

Academic Editor

Thank you for revising your manuscript according to the many helpful comments shared by reviewers. Congratulations and thank you for your novel and important work. Please look at the additional comment from R1, which you may want to consider. - Ruwan

Reviewer Comments (if any, and for reference):

Reviewer's Responses to Questions

**Comments to the Author**

1. If the authors have adequately addressed your comments raised in a previous round of review and you feel that this manuscript is now acceptable for publication, you may indicate that here to bypass the “Comments to the Author” section, enter your conflict of interest statement in the “Confidential to Editor” section, and submit your "Accept" recommendation.

Reviewer #1: (No Response)

2. Does this manuscript meet PLOS Global Public Health’s publication criteria? Is the manuscript technically sound, and do the data support the conclusions? The manuscript must describe methodologically and ethically rigorous research with conclusions that are appropriately drawn based on the data presented.

Reviewer #1: Yes

3. Has the statistical analysis been performed appropriately and rigorously?

Reviewer #1: Yes

4. Have the authors made all data underlying the findings in their manuscript fully available (please refer to the Data Availability Statement at the start of the manuscript PDF file)?

Reviewer #1: (No Response)

5. Is the manuscript presented in an intelligible fashion and written in standard English?

Reviewer #1: Yes

6. Review Comments to the Author

Reviewer #1: Thank you to the research team for the thorough responses to the comments. This will be a great addition to the toolkit of methods to document IHL violations. I have recommended the manuscript for acceptance, but I would suggest more qualification for one repeated claim in the manuscript: the conclusion that as facility size was not associated with damage, attacks were likely targeted (lines 34-35, 330-334, etc.). Lines 369-381 briefly address this claim, highlighting that larger buildings would be expected to suffer more attacks in the absence of other risk factors and offer two possible explanations: 1) intentional attacks on medical facilities, or 2) widespread, indiscriminate infrastructure damage. In a context/analysis with so many other risk factors/unobserved variables and such a high proportion of facilities damaged, and without analysis on the spatial distribution of damaged/undamaged health facilities in relation to conflict events, what was the basis to conclude that the explanation was intentional attacks over indiscriminate damage? I understand it's an IHL violation either way, but it remains unclear why the conclusion emphasizes one possible explanation over the other, especially as there are many viable other risk factors that may offer insight as to why small and large facilities experienced similar frequency of damage.

7. PLOS authors have the option to publish the peer review history of their article (what does this mean?). If published, this will include your full peer review and any attached files.

**Do you want your identity to be public for this peer review?** For information about this choice, including consent withdrawal, please see our Privacy Policy.

Reviewer #1: No
